# Study of Amine Functionalized Mesoporous Carbon as CO$_2$ Storage Materials

**Muhamad Faisal, Afif Zulfikar Pamungkas and Yuni Krisyuningsih Krisnandi** *

Solid Inorganic Framework Laboratory, Department of Chemistry, Faculty of Mathematics and Natural Science, Universitas Indonesia, Depok 16424, Indonesia; muhamad.faisal01@ui.ac.id (M.F.); afif.zulfikar@sci.ui.ac.id (A.Z.P.)
* Correspondence: yuni.krisnandi@sci.ui.ac.id; Tel.: +62-812-1856-7060

**Abstract:** Carbon sequestration via the carbon capture and storage (CCS) method is one of the most useful methods of lowering CO$_2$ emissions in the atmosphere. Ethylenediamine (EDA)- and triethylenetetramine (TETA)-modified mesoporous carbon (MC) has been successfully prepared as a CO$_2$ storage material. The effect of various concentrations of EDA or TETA added to MC, as well as activated carbon (AC), on their CO$_2$ adsorption capacity were investigated using high-purity CO$_2$ as a feed and a titration method to quantitatively measure the amount of adsorbed CO$_2$. The results showed that within 60 min adsorption time, MCEDA49 gave the highest CO$_2$ capacity adsorption (19.68 mmol/g), followed by MC-TETA30 (11.241 mol/g). The improvement of CO$_2$ adsorption capacity at low TETA loadings proved that the four amine functional groups in TETA gave an advantage to CO$_2$ adsorption. TETA-functionalized MC has the potential to be used as a CO$_2$ storage material at a low concentration. Therefore, it is relatively benign and friendly to the environment.

**Keywords:** CO$_2$ capture; adsorption; amine-based adsorbents; mesoporous carbon; triethylenetetramine; ethylenediamine





## 1. Introduction

Climate change and global warming, which are currently attracting a lot of attention, are caused by greenhouse emissions, such as carbon dioxide (CO$_2$), methane (CH$_4$), chlorofluorocarbons (CFC), ozone, and dinitrogen monoxide, into the atmosphere [1]. The effect of CO$_2$ is the primary cause of global warming because of the use of fossil fuels, which contributes about 98% of the world's energy needs [1]. Thus, the amount of CO$_2$ emitted is very great. The concentration of CO$_2$ in the atmosphere is gradually increasing—it already hit 409.8 ± 0.1 ppm in 2019 and this differed significantly from that in 1970, around 324 ppm [2]. In accordance with the Paris Agreement, the Indonesian government has promised to lower its greenhouse gas emissions about 29–41% by 2030 [3]. Carbon sequestration via carbon capture and storage (CCS) is one of the most useful methods used to lower greenhouse gas emissions, since it is technically and economically feasible [4]. The concept of CCS is to capture CO$_2$ released from industrial processes and then transport it to a storage area [5].

Among CCS, post combustion capture (PCC) has been the most frequently used CCS strategy [4]. PCC using a sorption-based process is very promising because it has high efficiency and selectivity and it is quite feasible [6,7]. The aqueous solvents of alkanolamine, such as monoethanolamine (MEA) and diethanolamine (DEA), have been commonly used as amine-based solvents for PCC. However, there are some disadvantages related to the aqueous absorption-based process, such as low absorption capacity, poor stability and the high regeneration cost [8]. These problems can be overcome by using porous adsorbents that have high surface area and pore volume, such as porous carbon materials, zeolites and metal organic frameworks (MOFs) [9–12]. However, the affinity of solid adsorbents to CO$_2$ is lower than that of the aqueous amine solution. Therefore, to improve its affinity to CO$_2$, the solid surface is modified by amine compounds [13]. Pinto et al. (2011) reported that

amine-functionalized clay that reacted with $CO_2$ had one strong [13]C NMR resonance at 164 ppm, derived from carbamate species, formed by the reaction of $CO_2$ with amine groups on the surface of the clay [14]. On the other hand, clay that reacted with $CO_2$ only displayed a single peak at 125 ppm, contributing to $CO_2$ being physically adsorbed [14]. Furthermore, to increase the amount of $CO_2$ that was adsorbed, the use of mesoporous carbon, which has large surface area and pore volume, could be advantageous in accommodating large amounts of impregnated amines [15]. Therefore, in this study, amine-group modified mesoporous carbon was prepared to be applied for the purpose of $CO_2$ capture.

Mesoporous material has the potential to be used as $CO_2$ capture material because it has a high surface area, good thermal/chemical stability and a surface that can be easily functionalized [9,16–18]. Wang et al. prepared mesoporous ZSM-5-functionalized tetraethylenepentamine (TEPA) and validated its higher $CO_2$ uptake than ZSM-5 without modification, with an optimum adsorption capacity of 1.80 mmol/g at 100 °C [19]. Nguyen et al. prepared mesoporous LTA zeolites with various mesopore sizes, and the $CO_2$ adsorption tests showed that the amount of $CO_2$ uptake after functionalizing LTA zeolites with (3-amino-propyl) trimethoxysilane (APTMS) significantly increased because aminosilanes grafted onto the mesoporous surface and the adsorption sites in LTA simultaneously contributed to the total $CO_2$ uptake [20]. Gómez-Pozuelo and co-workers modified a wide series of clays (montmorillonite, bentonite, saponite, sepiolite and palygorskite) with amine compounds using three different amine compounds—aminopropyl organosilanes, diethylenetriamine organosilanes and polyethyleneimine—using grafting, impregnation and double functionalization methods [13]. The results showed that sepiolite grafted with diethylenetriamine and palygorskite impregnated with polyethylenimine had the highest $CO_2$ uptake of 61.3 and 67.1 mg $CO_2$/g and were adsorbent, with an efficiency of 0.38 and 0.21 mol $CO_2$/mol N, respectively [13].

On the other hand, mesoporous carbon has the potential to be used as a $CO_2$ storage material because it features many advantages, including a large surface area, high stability and ease of surface functionalization [21]. However, physical adsorption makes mesoporous carbon temperature-dependent, and it has low selectivity [22]. For these reasons, the development of carbon-based materials to be used as $CO_2$ storage materials are mainly focused on improving the $CO_2$ adsorption capacity through surface modification [23]. Introducing amine functional groups into mesoporous carbon surfaces could overcome this, as many studies have proven that the $CO_2$ adsorption capacity of carbon-based materials increases significantly when functionalizing its surface with nitrogen functional groups [24–27]. Another study has been conducted by Peng et al., which tested the $CO_2$ adsorption capacity of chitosan-derived mesoporous carbon (CDMC)-modified pentaethylenehexamine (PEHA) [28]. The results showed that the impregnation of CDMC with PEHA significantly increases the $CO_2$ uptake, but as the PEHA loading increases, the $CO_2$ uptake decreases, because it has complicated effects on $CO_2$ adsorption capacity—the increase in the PEHA concentration offers more nitrogen functional groups to react with $CO_2$, but it also blocks the $CO_2$ diffusion into deep layers [28]. Activated carbon has also been used frequently as a $CO_2$ capture material because it is abundant, has a large surface area and excellent thermal/chemical stability and it does not require a lot of energy to regenerate [29].

A preliminary study by our groups used methyl diethanolamine (MDEA)-functionalized MC and it showed that the adsorption capacity increased as the concentration of MDEA increased and reached the maximum capacity at 2.63 mmol/g for MC-MDEA43 [30]. The results also showed that the adsorption capacity of MC-MDEA50 decreased to about 33% [30]. This shows that high concentration of amine compounds also has a complex effect on the $CO_2$ adsorption capacity.

In search of effective amine-functionalized carbon materials to be used as $CO_2$ storage materials, in this work, mesoporous carbon was synthesized, amine-functionalized with ethylenediamine (EDA) and triethylenetetramine (TETA), which were prepared through the wet impregnation method. Both amine compounds were smaller than PEHA and had

a lower number of $NH_2$-functional groups compared to PEHA. Samples of mesoporous carbon functionalized with different EDA or TETA concentrations were investigated for their $CO_2$ adsorption capacity. As a comparison, commercially-available activated carbon and its amine-modified derivatives were also tested for $CO_2$ adsorption. Based on the experimental results, the possibility of MCTETA to be used as more green and benign $CO_2$ capture material was discussed.

## 2. Materials and Methods

### 2.1. Experimental Materials

Phloroglucinol, Pluronic F-127 and triethylene-tetraamine (60%) were purchased from Sigma-Aldrich (St. Louis, Missouri, US). Ethylenediamine and ethanol 99.9% were purchased from Merck (Darmstadt, Germany), formaldehyde 37% was purchased from Smart-Lab Indonesia (South Tangerang, Indonesia) and HCl 37% was purchased from Ajax Finechem (Sydney, Australia). $CO_2$ (UHP 99.99%) was purchased from CV Retno Gas (Jakarta, Indonesia). AC technical grade, Carbomax® (Jakarta, Indonesia), was purchased from a local vendor.

### 2.2. Synthesis of Mesoporous Carbon

Mesoporous carbon was synthesized using the soft template method according to a modified procedure [31]. Mesoporous carbon was prepared using phloroglucinol, formaldehyde and poly(ethylene-oxide)-poly(propylene oxide) triblock copolymer (Pluronic F127) as carbon sources. Initially, phloroglucinol and Pluronic F127 were mixed with a 10:9 weight ratio of ethanol and water and then stirred at room temperature. After that, HCl 37% was slowly added into the mixture and stirred for an hour. Finally, formaldehyde 37% was added to the mixture. After 2 h, it was caramelized and separated into two layers. The bottom layer was then taken and stirred overnight. It turned into an elastic monolith. The monolith was then hydrothermally processed in an autoclave at 100 °C for 24 h. The resulting black powder was then carbonized in the tubular furnace under nitrogen gas flow with gradual heating from 100 °C to 850 °C. The resulting mesoporous carbon (labeled as MC) as well as the activated carbon (AC) were characterized using Fourier Transform Infrared (FTIR) spectroscopy, X-ray Diffraction (XRD), Energy Dispersive X-ray (EDX) Spectroscopy and Surface Area Analysis-nitrogen adsorption desorption measurements.

### 2.3. Modification of Carbon Materials with Amine-Functional Groups

As-synthesized mesoporous carbon (MC) and activated carbon (AC) were modified with amine-functional groups through the wet impregnation method. In a typical preparation, a desired amount of ethylenediamine (EDA) with a weight ratio to mesoporous carbon of 50% wt and 100% wt was dissolved in 10 mL ethanol, then MC or AC was dispersed to the solution, which was stirred further for 6 h. Ethanol was left to evaporate at room temperature overnight. Similar preparation method was applied to triethylenetetraamine (TETA)-modified MC and AC. The amount of amine compounds attached to the carbon materials was determined based on EDX data (available in Supplementary Materials). The as-prepared adsorbent was denoted as MC-EDAx/MC-TETAx or AC-EDAx/AC-TETAx, where x represents the weight percentage of EDA/TETA. Based on calculation of the EDX measurement and the titration method, the modified MC and AC were labeled as MCEDA30, MCEDA49, MCEDA100, MCTETA10, MCTETA30, MCTETA52, ACTETA14 and ACTETA21.

### 2.4. Characterization

The Fourier transform infrared (FTIR) spectra were recorded using a Bruker Alpha FTIR spectrophotometer (Amherst, MA, USA) at room temperature. Before the measurement, the adsorbent was prepared as a KBr pellet, and approximately 2 mg of the sample mixed with 40 mg KBr (ca 5 wt%) to make a thin and transparent KBr pellet. X-ray diffraction (XRD) was performed on a Rigaku Smartlab diffractometer with a Cu Kα beam

($\lambda$ = 1.54 Å) (Rigaku Corporation, Tokyo, Japan). The patterns were recorded at 2$\theta$ of 10°–90°. The $N_2$ adsorption-desorption isotherms were measured using Quantachrome QuadraWin 2000–16 (Quantachrome Instruments, Boynton Beach, FL, USA). The measurement was performed in liquid $N_2$ medium at 77.3 K. The specific surface area ($S_{BET}$) was calculated using the Brunaeur–Emmett–Teller (BET) method, whereas micro- and meso-porosity parameters were calculated using the *t*-plot method. The elemental composition was analyzed using EDX (FEI INSPECT F50, Shinagawa, Japan).

### 2.5. Adsorption of $CO_2$

The adsorption of $CO_2$ gas was conducted according to the procedure described in [32]. The adsorption of $CO_2$ gas was conducted in a chamber with dimensions of 8 × 3 × 3 cm containing 0.1 g of adsorbent. The adsorption was carried out with pure $CO_2$ (UHP 99.99%), at a flow rate of 50 mL/min and a pressure of 5 atm, controlled with a 0–100 mL scale on a Dwyer flowmeter (8% accuracy, Dwyer Instruments, Michigan City, MI, USA). The adsorption was performed for a certain time, which varied from 5 min to 60 min. The chamber was connected to a closed container filled with cold NaOH solution (100 mL, 0.00001 M, temperature ± 2 °C) to absorb the $CO_2$ that passed through and came out from the chamber. The $CO_2$ gas was then converted into an equivalent of sodium carbonate.

Determination of $CO_2$ concentration in the NaOH solution was carried through a titration method [33]. Ten milliliters of NaOH-$CO_2$ solution were titrated with 0.00001 M HCl solution. Titration to the first colorless phenolphthalein (PP) equivalence point neutralized the excess sodium hydroxide solutions and converted the sodium carbonate to sodium bicarbonate. Further titration to the second methyl orange equivalence point converted the sodium bicarbonate to water and carbon dioxide. The concentration of $CO_2$ dissolved in NaOH solution was calculated by the difference in milliliters between the first and second equivalence points, which is the method described in [33]. The amount of $CO_2$ adsorbed by carbon or amine-functionalized carbon materials was calculated from the mol difference between $CO_2$ from the blank experiment, i.e., without adsorbent, and the $CO_2$ that dissolved in the NaOH solution from each experiment. HCl was standardized using a tetraborate solution (0.00001 M) to validate the calculation of the $CO_2$ concentration in the NaOH solution. The standardization of the 0.00001 M NaOH solution was done with HCl standard solution. Calculation of the amount of adsorbed $CO_2$ was carried out following Equations (1)–(3) [32].

$$\text{Mol of } CO_2 \text{ passing through the adsorbent} = \text{Mol of } CO_2 \text{ in NaOH solution} \tag{1}$$

$$\text{Mol of } CO_2 \text{ in NaOH solution} = \frac{1}{2} \{\text{mol of NaOH} - (\text{M HCl} \times \Delta \text{V HCl})\} \tag{2}$$

$$\text{Mol of } CO_2 \text{ adsorbed} = \text{mol of } CO_2 \text{ (blank)} - \text{mol of } CO_2 \text{ passing through the adsorbent} \tag{3}$$

The experimental error of this measurement was 10.65%.

### 3. Results

#### 3.1. Adsorbent Characterization

3.1.1. X-ray Diffraction (XRD)

Figure 1 shows the powder XRD pattern of mesoporous carbon, confirming the identity of graphitic carbon. The material shows a strong diffraction peak at 2$\theta$ = 23°–26° and a weak diffraction peak at 43°–46°, corresponding to the diffraction of the (002) and (100) planes of the graphite structure, respectively [5–7].

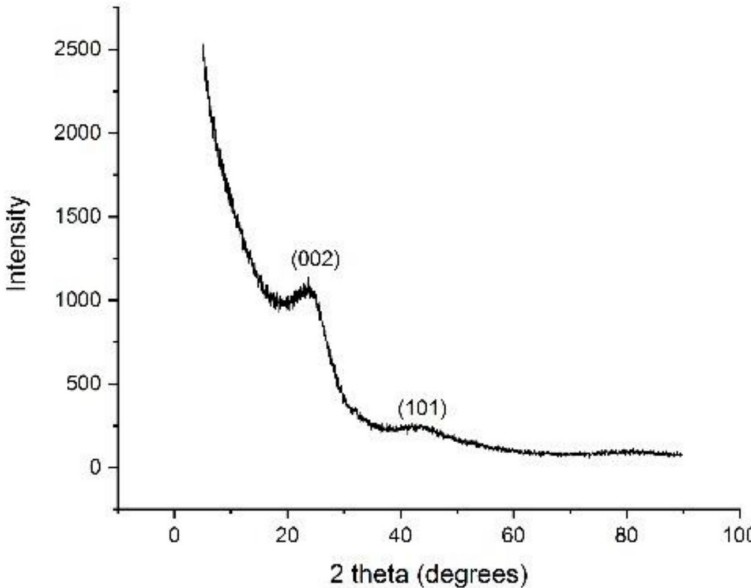

**Figure 1.** X-ray diffraction pattern of as-synthesized mesoporous carbon.

### 3.1.2. Fourier Transform Infrared Spectroscopy (FTIR)

For brevity, the FTIR spectra of mesoporous carbon before and after the carbonization process, as well as those of activated carbon, are not shown. They are available in Supplementary Materials. After the carbonization process, bands attributed to C-H stretching and bending vibration from formaldehyde and phloroglucinol [34] disappeared, showing the complete removal of most oxygen and hydrogen atoms of the precursors.

The amine-functionalized carbon was also characterized using FTIR. Functionalization of MC and AC with ethylenediamine (EDA) or triethylenetetramine (TETA) could be seen by the appearance of surface N-H functional groups. As shown in Figure 2, after modification, new bands at 1100–1250 cm$^{-1}$, 1500–1750 cm$^{-1}$ and 3250–3500 cm$^{-1}$ appeared, which were assigned to C-N stretching, N-H stretching and bending vibrations, respectively [35]. Furthermore, the intensity of these three bands increased as the amount of added TETA increased (Figure 2b,c). The difference between EDA- and TETA- modified MC and ACTETA spectra was that the bands around 1750–1250 cm$^{-1}$ of the modified MC were more visible, indicating a higher amount of amine compounds attached to the surface of MC than to the surface of AC. This is possible since the pore size of MC is larger than AC, as mentioned in Table 1.

**Table 1.** Textural parameters of mesoporous carbon and commercial activated carbon before and after amine-functionalization. $S_{BET}$, specific surface area, calculated using the Brunauer–Emmett–Teller (BET) method.

| Material | $S_{BET}$ [a] (m$^2$ g$^{-1}$) | $S_{ext}$ [b] (Mesopore Area) (m$^2$ g$^{-1}$) | $S_{mic}$ [b] (Micropore Area) (m$^2$ g$^{-1}$) | $V_{tot}$ [c] (Total Pore Volume) (cm$^3$ g$^{-1}$) | $V_{micro}$ [b] (Micropore Volume) (cm$^3$ g$^{-1}$) | $V_{meso}$ (Mesopore Volume) (cm$^3$ g$^{-1}$) |
|---|---|---|---|---|---|---|
| MC | 391.11 | 295.08 | 96.02 | 0.738 | 0.045 | 0.693 |
| MC-EDA49 | 290.29 | 262.40 | 27.89 | 0.637 | 0.01 | 0.627 |
| MC-EDA100 | 163.19 | 163.19 | 0 | 0.301 | 0 | 0.301 |
| MC-TETA30 | 161.30 | 161.30 | 0 | 0.407 | 0.06 | 0.347 |
| MC-TETA52 | 68.35 | 68.35 | 0 | 0.107 | 0 | 0.107 |
| AC | 518.90 | 268.20 | 250.70 | 0.491 | 0.234 | 0.257 |
| AC-TETA14 | 17.83 | 17.83 | 0 | 0.082 | 0.0067 | 0.0753 |
| AC-TETA21 | 17.27 | 17.27 | 0 | 0.079 | 0.0062 | 0.0728 |

[a]: Calculated using BET method; [b]: Calculated using t-plot method; [c]: Calculated at P/Po ~0.993.1.3. BET Surface Area Analysis.

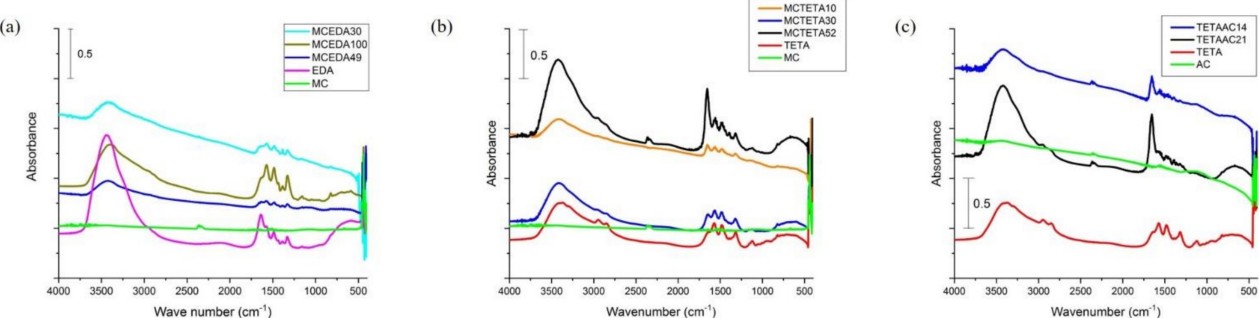

**Figure 2.** FTIR spectra of (**a**) ethylenediamine (EDA-) and (**b**) triethylenetetramine (TETA-)functionalized mesoporous carbon (MC), and (**c**) TETA-functionalized activated carbon (AC).

The textural properties of as-synthesized MC and commercial AC before and after functionalization with EDA or TETA were measured using $N_2$ adsorption-desorption measurements, and the results are shown in Figure 3. The as-synthesized mesoporous carbon (Figure 3a) exhibited a type-IV isotherm with type II hysteresis loops, which indicates the presence of mesopores [36,37]. Functionalization of as-synthesized mesoporous carbon with EDA or TETA did not change its mesoporous characteristics, although the intensity of the curves was lower. Meanwhile, the isotherms of AC and its amine derivatives (Figure 3b) displayed a type-II profile of microporous materials and the intensity of the curves was much lower.

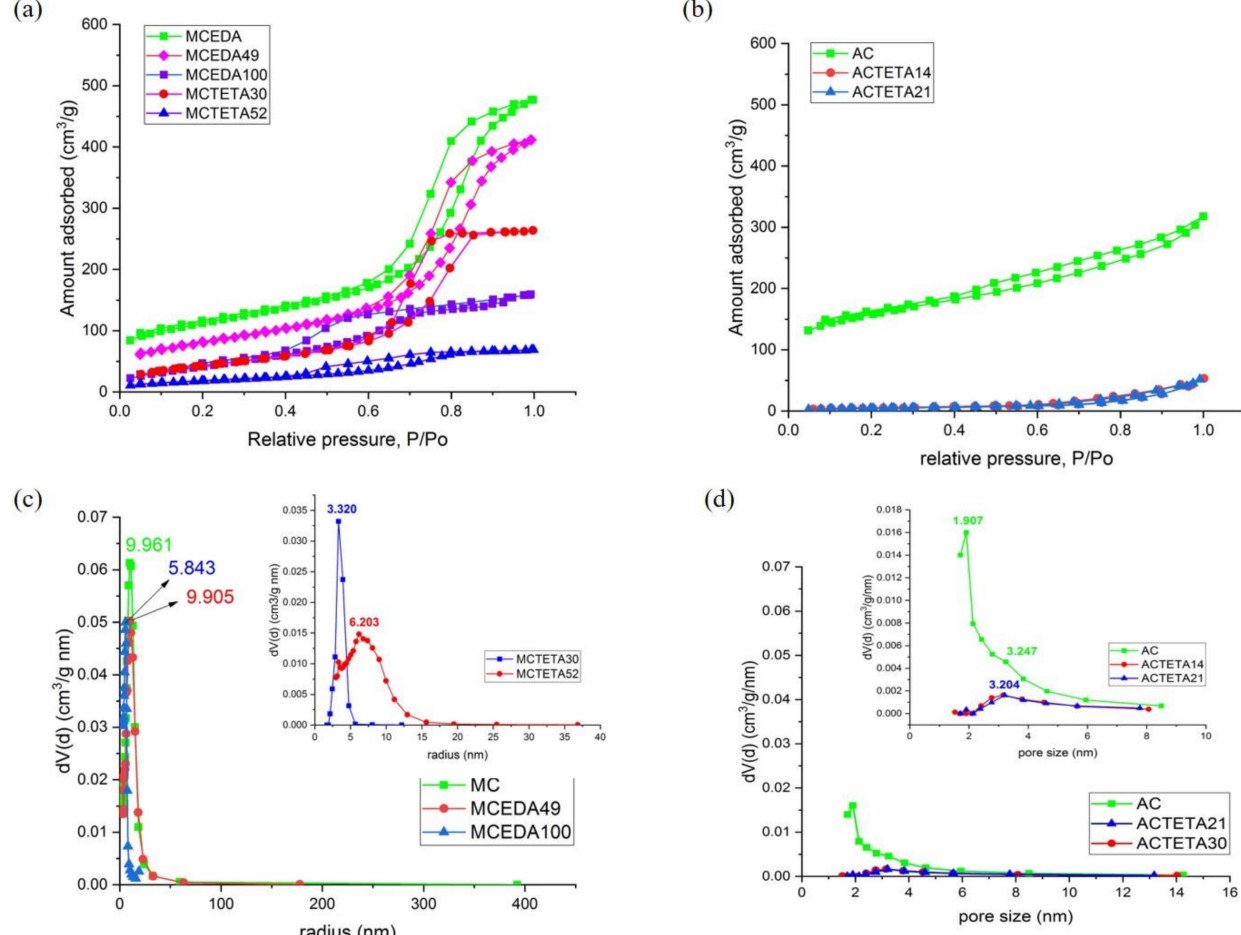

**Figure 3.** $N_2$ adsorption-desorption isotherm of amine-functionalized MC (**a**) and AC (**b**) and pore-size distribution of amine-functionalized MC (**c**) and AC (**d**).

The graphs displaying pore-size distribution in Figure 3b,c were obtained using Barrett–Joyner–Halenda (BJH) desorption curves. The pore-size distribution of as-synthesized mesoporous carbon showed a single peak at around 9.96 nm, confirming its mesoporosity [38]. In MCEDA-49, the pore size distribution was unchanged although the intensity decreased, whereas in MCEDA-100, the distribution was shifted to a lower pore-size distribution, indicating that pores were filled with EDA. Functionalization of MC and AC with TETA caused the specific surface area to decrease, which is supported by their BJH desorption curves, in which the distribution shifted to a smaller pore-size with much lower intensity. These results have demonstrated that the pore filling by TETA could block the pores, especially in microporous AC [39]. The textural properties of both carbon materials (MC and AC) and their amine-functionalized derivatives are summarized in Table 1.

### 3.1.3. $CO_2$ Adsorption Test

The effect of amine functional groups in amine-functionalized mesoporous carbon on its $CO_2$ adsorption capacity was studied first using ethylenediamine (EDA), which has two amine functional groups. Figure 4a shows the $CO_2$ adsorption capacity of mesoporous carbon loaded with EDA, labelled as MC-EDA30, MC-EDA49 and MC-EDA100. It was shown that the $CO_2$ adsorption in MCEDA30 reached the highest capacity of 29.95 mmol/g in 15 min adsorption, whereas MCEDA49 and MCEDA100 showed only 6.78 mmol/g and 2.34 mmol/g, respectively. However, after 30 min the adsorption capacity of MCEDA30 was dramatically decreased and reached the lowest value of 0.43 mmol/g within 60 min adsorption. On the other hand, for MC-EDA49, the $CO_2$ adsorption steadily increased along the $CO_2$ flowtime for 60 min, reaching 19.68 mmol/g. The adsorption in MCEDA100 was still inferior compared to that of pure MC.

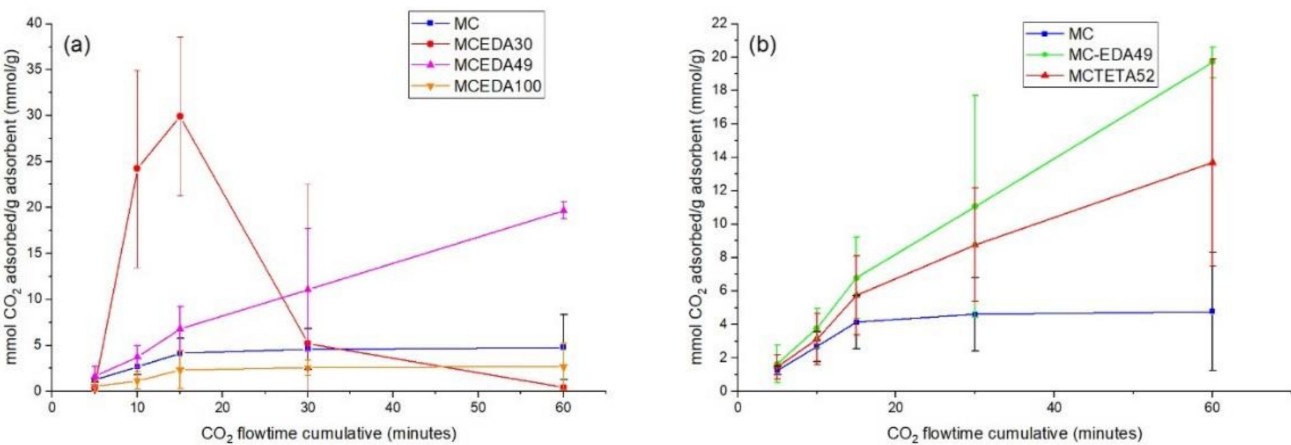

**Figure 4.** $CO_2$ adsorption capacity of MC and (**a**) MC-EDA; (**b**) MC-EDA50 and MC-TETA50, at pressure (P) of 5 atm at room temperature and with a $CO_2$ flow of 50 mL/min.

In general, a higher concentration of EDA provides more amine functional groups to readily react with $CO_2$, which is beneficial for $CO_2$ adsorption. However, as listed in Table 1, in MC-EDA100, a higher amount of EDA significantly decreased the $v_{meso}$ (volume of mesopore), which may lead to hindrance of the diffusion of $CO_2$ into the pores. The anomaly adsorption activity shown by MCEDA30 within 60 min $CO_2$ flowtime could be caused by two factors. First, the pores of MCEDA30 could become deactivated after 15 min, so more $CO_2$ flowing through was not picked up anymore. Second, the interaction between $CO_2$ and the amine groups in EDA that immobilized on the surface of mesoporous carbon was rather weak in stabilizing the $CO_2$. The reason why the $CO_2$ adsorption capacity of MCEDA30 after 60 min was below that of pristine MC is still not clear. Further characterization on MCEDA30 needs to be carried out to understand its adsorption activity towards $CO_2$.

However, since in this work, the focus is on studying the use of TETA as the functional group to improve the adsorption capacity of carbon materials, the MC surface was then modified with TETA. The aimed loading of 10, 30 and 50 wt% gave three MC adsorbents—MCTETA10, MCTETA30 and MCTETA52, respectively.

Figure 4b shows the $CO_2$ adsorption capacity of MC-EDA49 and MC-TETA52, respectively. Although the $CO_2$ adsorption capacity of MCEDA49 significantly increased during the 60 min adsorption, the $CO_2$ adsorption capacity of MC-TETA52 also increased, although it was less significant than that of MC-EDA49. Although the use of TETA provides more amine functional groups to readily react with $CO_2$ than EDA, the increased amount of amine groups could block the diffusion of $CO_2$ into the pores of MC. This could be a drawback for $CO_2$ adsorption. Therefore, the $CO_2$ adsorption in TETA-modified MCs lower than 50 wt% TETA (MC-TETA10 and MC-TETA30) was studied further.

Figure 5a shows the $CO_2$ adsorption capacity of MCTETA10 and MC-TETA30 in comparison with MC-TETA52. It shows that the $CO_2$ adsorption capacity of MCTETA10 is similar to the parent MC, indicating that the low concentration of TETA had no significant effect on its adsorption capacity. Interestingly, the $CO_2$ adsorbed in MC-TETA30 increased substantially, more than the amount of $CO_2$ adsorbed in MCTETA50. In addition, no deactivation occurred, as was found with $CO_2$ adsorption on MCEDA30. This result suggests that 30 wt% is the optimum TETA loading since the $CO_2$ adsorption experiences less pore blocking. Thus, the four amine functional groups in TETA benefit $CO_2$ adsorption capacity, and the diffusion barrier could be avoided. Based on this experiment, the highest $CO_2$ adsorption capacity was given by MCTETA30.

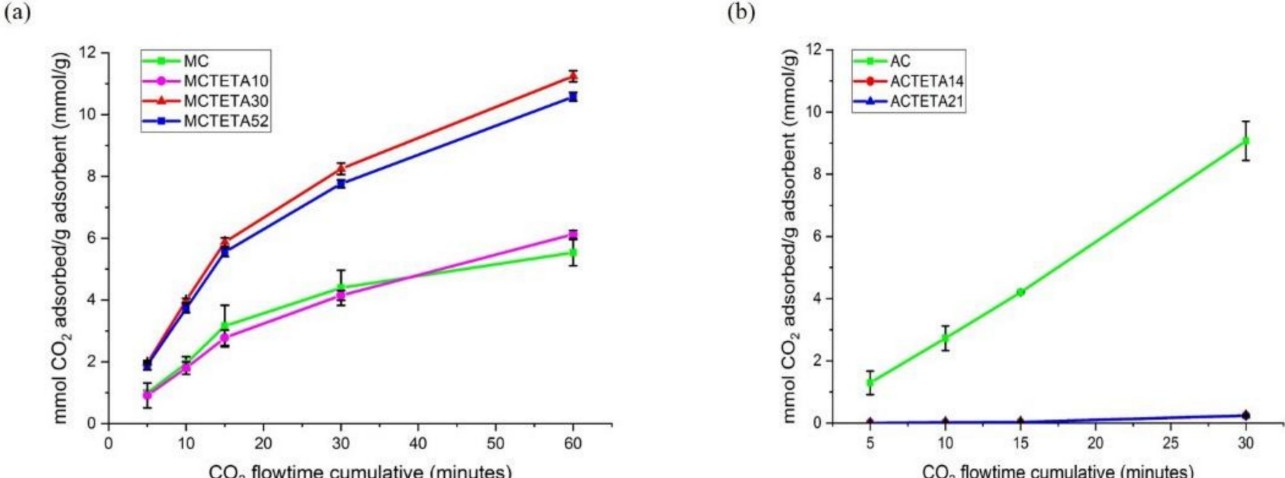

**Figure 5.** $CO_2$ adsorption curve for TETA-functionalized (**a**) MC and (**b**) AC at pressure (P) of 5 atm at room temperature and with a $CO_2$ flow of 50 mL/min.

The $CO_2$ adsorption capacity of MCTETA was then compared with the commercially available activated carbon (AC) and its TETA-modified derivatives (ACTETA14 and ACTETA21). It can be seen in Figure 5b that in 30 min $CO_2$ flowtime, AC showed a higher $CO_2$ adsorption capacity than MC. It is reported that $CO_2$ can also be adsorbed on the surface of graphitic carbon through physisorption of $CO_2$, and it is heavily dependent on the physicochemical properties of the materials [39]. As shown by the FTIR spectra in Figure 2, the surface of AC has more functional groups on the surface compared to MC. In addition, the surface area of AC is higher than that of MC. On the other hand, functionalization of AC with TETA decreases its $CO_2$ adsorption capacity, because their surface area and total pore volume decrease significantly. This indicates that the functionalization of TETA to the surface of AC may block its pores and leave limited access for $CO_2$. This was also reported by Houshmand et al. in the $CO_2$ adsorption on diethylenetriamine (DETA)-modified AC [39].

## 4. Discussion

The results presented above show that mesoporous carbon was successfully synthesized from Pluronic F-127, phloroglucinol and formaldehyde. The carbonization process, which involves the removal of most oxygen and hydrogen atoms of the carbon sources mentioned before, plays an important role in the preparation of graphitic mesoporous carbon. The functionalization of the surface of as-synthesized mesoporous carbon with amine compounds has a significant effect on its $CO_2$ adsorption capacity.

Functional groups that exist on the surface of carbon materials, i.e., amines and hydroxyl groups, play important roles in the $CO_2$ adsorption reaction. In MCEDA, the primary amine of EDA ($RNH_2$) reacts with $CO_2$ to form carbamate ions, as shown in reaction (R1) and as illustrated in Figure 6. In MCTETA, both primary and secondary amines ($RNH_2$ and $R_2NH$, respectively) react with $CO_2$, as illustrated by reaction (R2) and as illustrated in Figure 7. On the other hand, the hydroxyl functional groups in the activated carbon could react with $CO_2$ to form bicarbonate ions by reaction (R3) and as illustrated in Figure 8 [40] This explains why both MC and AC already have $CO_2$ adsorption capacity prior to modification.

On the other hand, the $CO_2$ adsorption capacity of TETA-modified activated carbon (AC) is lower than that of TETA-modified MC, indicating that the functionalization of AC with amine-functional groups may block its pores and leave limited access for $CO_2$ to pass through due to decreases in its surface area and pore volume.

$$CO_2 + 2RNH_2 \longrightarrow RNH_3^+ + RNHCOO^- \tag{R1}$$

$$CO_2 + 2R_1R_2NH \longrightarrow R_1R_2NH_2^+ + R_1R_2NCOO^- \tag{R2}$$

$$CO_2 + OH^- \longrightarrow HCO_3^- \tag{R3}$$

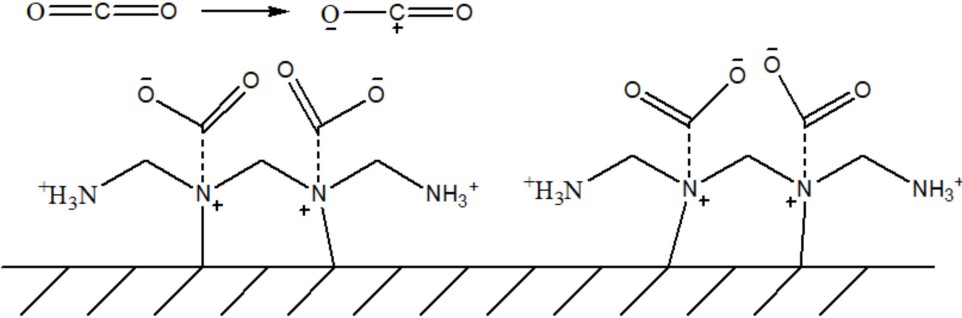

**Figure 6.** Proposed reaction mechanism between EDA and $CO_2$ at the surface of amine-functionalized mesoporous carbon.

**Figure 7.** Proposed reaction mechanism between TETA and $CO_2$ at the surface of mesoporous carbon.

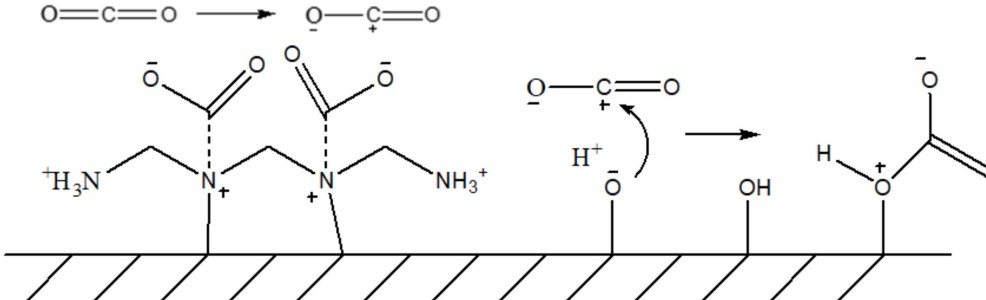

**Figure 8.** Proposed reaction mechanism between TETA and hydroxide functional groups of AC and $CO_2$.

Table 2 summarizes several works by other groups discussing amine-functionalized materials for capturing $CO_2$ [2,15,19,32,41–45]. The advantages of using mesoporous carbon as the support material for capturing $CO_2$ are it has a large surface area to accommodate a large amount of amine functional groups that contact with $CO_2$, and it does not require a complicated synthesis process [46].

**Table 2.** Amine-modified materials tested for $CO_2$ adsorption from the literature and this work.

| Material | Temperature (°C) | Adsorption Capacity (mmol/g) | Quantification Method | Reference |
|---|---|---|---|---|
| SBA15-HBP(DETA) | 30 | 0.41 | Gas Chromatography | [41] |
| SBA15-HBP(TETA) | 30 | 0.68 | Gas Chromatography | [41] |
| ZSM-5(TEPA) | 100 | 1.8 | Gravimetric | [19] |
| Zn/Co-ZIF(PEI40) | 25 | 1.82 | Microbalance | [43] |
| SBA15-HBP(TEPA) | 30 | 2.11 | Gas Chromatography | [41] |
| MCM-41(TEPA60) | 70 | 2.45 | Gas Chromatography | [2] |
| MC-PEI70 | 25 | 2.82 | Gas Chromatography | [44] |
| SBA-15-PEI(600) | 30 | 3.2 | Gas Chromatography | [41] |
| MCM-41-APTS30-TEPA40 | 70 | 3.45 | Gas Chromatography | [2] |
| MIL101-(Cr)-PEI70 | room temperature | 3.81 | Gas Chromatography | [42] |
| MOF-5 | room temperature | 5.46 | Titration * | [32] |
| MC-PEI65 | 75 | 4.12 | Gas Chromatography | [15] |
| MC-TETA30 | room temperature | 11.24 | Titration * | This work |
| MC-EDA49 | room temperature | 19.68 | Titration * | This work |
| [Cu3(BTC)2] | room temperature | 234.26 | Titration * | [45] |

* Concentration of $CO_2$ used was 99.99% (UHP), whereas concentrations in other works were 10%–15%.

Although in this reaction MCEDA49 had the highest $CO_2$ adsorption capacity, the use of MCTETA materials also showed promising results, especially MCTETA30. Therefore, TETA can be used as an alternative to modify mesoporous carbon, because the amount of TETA needed is lower than EDA. This is one example of implementation in green chemistry and is aligned to the Sustainable Development Goals, especially No.12: Responsible Consumption and Production [47].

**5. Conclusions**

The carbon materials—mesoporous and activated carbon—were modified with amine functional groups ethylenediamine and triethylenetetramine. MCEDA30 showed the best $CO_2$ adsorption capacity within a 15 min flow time, but at a longer time, MCEDA49 displayed a steady increase in $CO_2$ adsorption. Modification of MC with low TETA loadings has been shown to cause less pore-blocking and diffusion barriers; thus, its four amine functional groups are beneficial for $CO_2$ adsorption capacity. The results also showed that when TETA was chosen, the highest $CO_2$ adsorption capacity was shown by MCTETA30.

A high concentration of amine compounds can block the pores of carbon materials, which could hinder the interaction between carbon dioxide and amine functional groups. To conclude, this study suggests that TETA-functionalized MC has the potential to be used as a $CO_2$ storage material, as it can be used at a low concentration, and is thus more benign and friendly to the environment.

**Supplementary Materials:** The following are available online at https://www.mdpi.com/2227-9717/9/3/456/s1, Figure S1: FTIR spectra of MC, Table S1. EDX for MCEDA49, Table S2. Name of adsorbent materials.

**Author Contributions:** Conceptualization, Y.K.K. and A.Z.P. and M.F.; methodology, Y.K.K., A.Z.P. and, M.F.; software, M.F.; validation, Y.K.K.; formal analysis, Y.K.K., A.Z.P. and M.F.; investigation, M.F. and A.Z.P.; resources, M.F.; data curation, Y.K.K.; writing—original draft preparation, M.F., A.Z.P. and Y.K.K.; writing—review and editing, A.Z.P., M.F. and Y.K.K.; visualization using ChemDraw, M.F.; supervision, Y.K.K.; project administration, Y.K.K.; funding acquisition, Y.K.K. All authors have read and agreed to the published version of the manuscript.

**Funding:** This research was funded by Universitas Indonesia through UI Research Grant No. NKB-1661/UN2.RST/HKP.05.00/2020.

**Institutional Review Board Statement:** Not applicable, as the study did not involve humans or animals.

**Informed Consent Statement:** Not applicable, as this study did not involve humans.

**Data Availability Statement:** Not applicable.

**Acknowledgments:** The authors express gratitude to Adel Fisli at BATAN for access to BET measurements.

**Conflicts of Interest:** The authors declare no conflict of interest.

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
