# Peer review of "Study of Amine Functionalized Mesoporous Carbon as CO2 Storage Materials"

_processes, doi:10.3390/pr9030456_

Round 1

Reviewer 1 Report

Review processes-1074666

Title: Study of Amine Functionalized Mesoporous Carbon as CO2 Storage Materials

The present paper describes the preparation of mesoporous carbon materials and its application as solid adsorbents in CO2 capture. The topic is relevant and the work is interesting. However, in my opinion the paper needs a lot of improvements. Here are some comments in order to improve the quality of the work.

1-Introduction Section

The English writing needs a carefully revision. Here are a few examples:

Line 4: “The effect of CO2 are the primary cause of global warming”. Is instead of are

Line 5: “Thus, the amount of CO2 emitted are much greater.” Is instead of are. Much greater than what?

Line 6: “The concentration of CO2 in the atmosphere is keep increasing gradually”. You can delete keep

Line 13: “post combustion capture (PCC) has been the most CCS strategy that frequently used”. has been he CCS strategy most frequently used.

Line 14: “PCC using sorption-based process is very promising because it has high efficiency and selectivity and it is inexpensive” I am not aware of any CO2 capture process that is inexpensive, please correct the information.

There is much more English writing that has to be corrected in this paper. The paper has to be revised by someone fluent in English.

2- Materials preparation

The authors don’t describe how are the materials are recovered after the wet impregnation. What happens to ethanol? The materials are dried at which temperature? And for how long?

Throughout the text the authors refer the materials as 50% loaded or 100% loaded. However, these values are in fact not the loading value, but the quantity of amine used during the impregnation method. This misleads the reader. For example, in page 8, 2nd paragraph.: “This result suggests 30 wt% is the optimum TETA loading.” This isn’t correct, the authors have to clarify this information.  

3- Regarding adsorption experiments:

Page4, section 2.5

-The authors state that: “The adsorption was performed for 10 min” – the authors have to describe the experiments in a general mode. The time of adsorption was different for each experiment. I believed it has varied from 5 to 60 minutes. The authors have to correct the information.

-The authors should explain in the paper why did they use pure CO2 for the adsoption experiments, if post combustion capture fue gas has 15%CO2 maximum.

-The authors should give a more complete description of the experiments. For example, which were the dimensions of the chamber used? specify at which purity CO2 was used in the adsorption experiments? How was the CO2 flow measured and controlled?

-I have some questions about the error of the experimental method used. For example, for blank experiment with MC. After 10 minutes the CO2 that entered the system was 4.5 grams and the CO2 that leaved the chamber was 4.4868 g. Do the authors have the precision needed to quantify such a low difference. How precise are measurements? How does the authors measure the flow of CO2 that enters the chamber?  Which is the equipment used to measure this flow?

-Also, the authors should present the error bar of each experiment. How many times was the same experience repeated and what was the % of error. How reproducible are these experiments?

Besides reference 32 (by the same authors) and 45, the authors should give more references of papers using the titration method to calculate CO2 adsorption capacity on solid adsorbents. If there are no more references, the authors have to validate their method by measuring a known sample and comparing with other authors. I have a lot of reservations about the precision of these results.

4- Regarding Figure 5, 6 and 7:

-Something is wrong with data from Figures 5 and 6. If we compare the data for the material MC orange in figure 5 and blue in figure 6, there is a clear difference between values that should be the same. Because it is the same experiment.

-Furthermore, the authors should use the same colour for the same material in different figures. This will help the reader to analyse the results.

-Also, the experimental conditions as temperature, pressure and CO2 flow should appear in the figure caption.

-Regarding Figure 7b, the yy axis start before 0???

5- Regarding table 5

There is another important factor that the authors absolutely have to add to this table which is the CO2 concentration in the inlet flow. Most of the works that appear in the same table use CO2 concentrations between 10-15%. In this work pure CO2 was used. And this is relevant information that has to be in the table.

Decision:

In my opinion this paper needs a lot of improvements and is not ready to be submitted to a scientific journal.

Author Response

Dear Reviewer 1

Thank you for the valuable comments and suggestions given on our manuscript. We have revised the manuscript following them.
Please see the attachment.

Reviewer 2 Report

The paper includes significant work concerning a subject of interest, however, the results obtained are not conclusive, wel supported by the results. The research is not completely well designed. At least an additional experiment with MC functionalized with EDA would be needed. The paper could be published after major revision. These are the main drawbacks found:

-          MC-TETA30 was chosen as the best sample for CO2 adsorption. It is reported like this both in the abstract:  “The highest CO2 adsorption capacity was given by MC-TETA30 within 60 min adsorption (11.241 mol)”, in the conclusions, as well as in the whole manuscript. However, this is probably like this only because MC-EDA30 was not tested. When MC samples functionalized with a similar proportion of EDA and TETA were tested, namely 50, MC-EDA showed a significant superior performance (Figure 6, Table2).  As the authors state, this could be attributed to its higher BET surface area and pore volume. In consequence, to verify which amine is a better choice it is mandatory to test at least MC-EDA30, thought it would be better to have experiments with the same proportions used for MC-TETA, 50, 30 and 10.  

-          It is not so clear, as the authors say, that MC-TETA show stronger peaks of amine functional groups bands at 1750-1250 cm-1 than AC-TETA (Figure 3), that would confirm the greater interaction of TETA with the support. ¿Was the same amount of sample used? Moreover, if amine is blocking the pores, the amount of amine desorbed could not be related to the strength of interaction. The strength of the interaction could be analysed by performing temperature programed desorption experiments.

-          It seems that there is some change in the species adsorbed when comparing the patterns of pure TETA with that adsorbed on carbon, the peak with the maxima intensity changes, ¿Is it possible that some species other than TETA could be formed by reaction with carbon surface?

-          It make not sense to include N2 adsorption results in CO2 adsorption section. As the former are characterization results, they should be include in 3.1. Adsorbent characterization section along with the other characterization results.

-          The discussion, conclusions and abstract should be modified according to the new experiments performed.

-          There are several grammar errors in the manuscript, for example:

                             Page 1, the amount….. are more

                               Page 2, Nguyen et al. has been prepared…

Page 2, various mesopore size, and

Page2, results shows…

Author Response

Dear Reviewer 2

Thank you for the valuable comments and suggestions given on our manuscript. We have revised the manuscript following them.
Please see the attachment.

Reviewer 3 Report

The presented work describes the results concerning amine functionalized mesoporous carbon as CO2 sorbents. The results are interestig - high CO2 adsorption capacities were observed. However, there are a few things that need to be improved before the manuscript is consider for the publication:

  1. The abstract is not clear, therefore it is suggested to reorganize it and improve the language.
  2. How many times have adsorption experiments been repeated? The CO2 adsorption capacity (11.241 mol) is given to three decimal places, which would indicate a multiple repetition of the experiment.
  3. CO2 adsorption capacity in Abstract should be given in mol/g.
  4. I am not sure if the reference [45] to the last entry in Table 2 ([Cu3(BTC)2]) is correct.
  5. Authors concluded in Abstract that MC-TETA30 showed the highest CO2 adsorption capacity (11.241 mol/g) while in Table 2, for comparison with the literature data, MC-TETA50 (11.24 mmol/g) and MC-EDA50 (19.68 mmol/g) were selected. It is a bit confusing. Why MC-EDA50 was not selected as the best sorbent for which the highest adsorption capacity was observed?
  6. The diagram with the proposed mechanism is illegible.
  7. Extensive editing of English language and style is required.

Author Response

Dear Reviewer 3

Thank you for the valuable comments and suggestions given on our manuscript. We have revised the manuscript following them.
Please see the attachment.

Round 2

Reviewer 2 Report

If MCEDA reached a significant higher value of CO2 adsorption at 15 min than MC-TETA, it is not proved as the authors claim that the improvement of CO2 adsorption capacity at low loadings for TETA come from the four amine fuctional groups. The marked drop of the adsorption observed for MCEDA appears to be like a kind of deactivation which relation with the number of amine groups must be investigated. i am sorry but in the current manuscript, the conclusions are not well supported by the results and, in my opinion, further experiments and results analysis is needed before it can be published.

Author Response

Dear 2nd Reviewer,

Kind regards.

Round 3

Reviewer 2 Report

Changes included in the manuscript show more accurately the performance of the materials and the conclusions are not so radical and are better supported by the results.